# Revisiting the Conceptual Terrains of the Right to Accessibility in India: The Role of Judicial Enforcement

**Sanjay Jain [1],\* and Malika Jain [2]**

[1]  National Law School of India University, Bengaluru 560072, India
[2]  School of Law, MIT ADT University, Pune 412201, India; malika.jain@mituniversity.edu.in
\*  Correspondence: malikajain.09@gmail.com

**Abstract:** The main objective of this paper is to critically reflect on the right to accessibility of persons with disabilities in India, with special focus on the context of public streets and environments. The paper draws on work carried out during the India-related part of the Inclusive Public Space Project, as well as judicial pronouncements, and the norms evolved by India as a party to the United Nations Convention on the Rights of Persons with Disabilities. In this paper, we briefly set out competing conceptions of accessibility and evaluate its constitutional and statutory manifestations. Further, the measures undertaken by the government in the form of policies, guidelines and campaigns to ensure accessibility in the Indian socio-political infrastructure are also assessed. The same is followed by the analysis of significant judicial precedents of the Supreme Court and High Courts on different aspects of the right to accessibility, to demonstrate how the courts have spearheaded various structural enhancements in the accessibility regime in India. The paper is concluded by drawing attention to the need for greater synchronicity in the adjudication and implementation of the principle of accessibility.

**Keywords:** Rights of Persons with Disabilities Act 2016; UN Convention on the Rights of Persons with Disabilities; Persons with Disabilities Act 1995; right to accessibility; street accessibility; environmental accessibility

## 1. Introduction

The United Nations Convention on the Rights of Persons with Disabilities (UNCRPD) is the first major international human rights instrument to include a dedicated provision on accessibility and recognise its importance as a prime pre-condition of human rights. Close consideration of this treaty also establishes that it advocates for a transformative conception of accessibility.

While the right to accessibility of persons with disabilities on the international level has been the subject of numerous scholarly writings, considerably less attention has been given to the right in the context of the Indian legal framework, particularly with a focus on pedestrian environments. This paper addresses this gap. It also advances understanding of how the right to accessibility has been recognised and implemented through government and legislative initiatives, and by the courts in India. It highlights and critiques the active role which judges have played, and could play in the future, in driving forward accessibility on the ground. In this respect, the article makes a valuable contribution to the literature on the role of the judiciary in India. We also draw attention to the ongoing implementation and enforcement challenges and, further, we expose, and critique, the conceptions of disability and accessibility that underpin relevant laws, policies and judgements.

The article draws on and further develops research carried out by the first author as part of the research project, Inclusive Public Space: Law, Universality and Difference in the Accessibility of Streets. This took the form of a report on relevant law and policy developments in India prior to 2022. Although interview data were also collected in India as part of this project, those data are not presented in this paper.

A few clarifications are in order. First, the question of whether accessibility, as an overarching principle underlying the UNCRPD, is a right or an obligation is beyond the scope of this paper. Suffice to say that we regard accessibility as a right coupled with an obligation in a non-Hohfeldian sense; that is to say, it comes into play as an enforceable right, an overarching principle, and a standalone obligation. Second, we do not maintain any dogma about terminology and use "persons with disabilities" and "disabled persons" synonymously.

The article is divided into four main sections. In Section 2, we briefly set out the competing conceptions of accessibility, which serve as the basis of this paper. In Section 3, we provide a bird's-eye view of the constitutional and statutory adaptations of the principle of accessibility and measures undertaken by the government—in the form of policies, guidelines and campaigns—to ensure accessibility in the Indian socio-political infrastructure. Section 4 analyses significant judicial precedents of the Supreme Court and High Courts on the right to accessibility, with a view to evaluating the extent to which courts have spearheaded various structural enhancements in the Indian accessibility regime. Sections 5 and 6 pull the threads of the earlier discussion together. This discussion is followed by our conclusion, which focuses on the need for synchronicity in the adjudication and implementation of accessibility.

## 2. Competing Conceptions of Accessibility

In attempting to contextualise the notion of accessibility, reference may be made to the conception of "embodied access", as evolved by Titchkosky. She observes "[g]iven that questions of access can arise for anyone, at any time, and anywhere for innumerable reasons, access is a way people have of relating to the ways they are embodied as beings in the particular places where they find themselves" (Titchkosky 2011, pp. 3–4). Giving an expansive meaning to "embodied", she adds:

> all the ways one has to sense, feel, and move in the world, as these are mediated by the interests of social environments, including race, class, gender, and sexual orientation... Access not only needs to be sought out and fought for, legally secured, physically measured, and politically protected, it also needs to be understood—as a complex form of perception that organizes socio-political relations between people in social space. (Titchkosky 2011, pp. 3–4)

Similarly, Hendren observes that "accessibility in the broadest sense is a reminder that the built environment and all its structures are the products of human decisions" (Hendren 2020, p. 71). Thus, to transmit this message in the domain of roads and streets, it is arguable that navigational spaces must be designed inclusively—to cater to the needs of not only able-bodied people, but also people with a range of body/mind traits and circumstances.

In order to make the built environment accessible, not only physical, but also attitudinal barriers need to be eliminated. We can turn to activists like Nash to develop an ameliorative notion of advocacy. In this pursuit, we must take stock of the life journeys of successful role models and the impact they have left on society due to their tenacity and strong desire to move forward. Accessibility, apart from being an intrinsic public good, has an instrumental role in promoting and fostering physical and mental disabilities as distinct markers of identity. A positive and transformative approach towards the creation of an inclusive egalitarian and pluriversal environment of navigation has the potential to increase the productivity of all members of society as co-equal agents. It would also effectively dismantle stereotypes and stigma around the abilities and talents of persons with disabilities (Goffman 1986). The feeling of being a part of society as a co-player, apart from boosting confidence in oneself, would also instill a sense of responsibility to contribute meaningfully to all spheres. To put this concretely, accessible roads and streets would enable disabled people to freely navigate and approach places that are important in their lives, thereby enhancing their well-being and their opportunities to participate. Uninhibited navigation of streets would allow persons with disabilities to exercise their free choice and self-determination. Unimpeded streets would also be indicative of a transformation in the

consciousness of the mainstream of society—signifying a sea change in societal perceptions of the worth, potential and prospects of persons with disabilities.

Understood expansively, accessibility is conceived of broadly as a pre-condition for and part of development, as well as "for persons with disabilities to live independently and participate fully and equally in society" (Accessibility and Development: Mainstreaming Disability in the Post-2015 Development Agenda 2013). Accessibility, on this view, is an enabler right and is also a catalyst for the activation of other human rights, such as the right to access streets, to use public buildings and surf the internet. Accessibility is intricately enmeshed and inherent in the conception and realisation of equality—the enforcement of equality without accessibility is not consistent with the human rights standards and norms evolved by the UN for persons with disabilities. Accessibility as a precondition, rather than as a part of the post-facto compliance package, provides an impetus for the greater inclusion of and participation by persons with disabilities in the mainstream of society as equal agents and citizens.

In contrast with this progressive view of accessibility, its conventional and ableist conceptions entail regarding it as something demanding perfunctory measures, often introduced retroactively as part of assistance and rehabilitation initiatives and limited by the bounds of the economic capacity of the state and other relevant authorities. On this view, accessibility is neither perceived as a rights enabler nor as a catalyst for the activation of other rights. Rather, it is seen as a part of the rehabilitation programme of the State, through which it renders assistance and helps people with disabilities to remain as close to the community as possible.

In accordance with Foucault's analysis of Normality, it can be contended that, from this, less progressive perspective, accessibility initiatives represent programmes of embedding "normality" into the lifestyles of persons with disabilities, rather than serving as inclusive and transformative measures (Tremain 2015). Far from being neutral, normality is inextricably enmeshed with "societal, structural power" (Persson et al. 2015). This analysis can be deployed to challenge and contest the altruism associated with the accessibility regime and may also provide room for alternative paradigms. Persson et al. neatly capture this point by observing that "[f]rom this discursive perspective, one could argue that policy building around accessibility is as much a question of consolidating a societal structure, as it is a question of including groups previously excluded from parts of society" (Persson et al. 2015). Thus, traditional or conventional accounts of accessibility do not regard it as a right, but instead relegate it to a privilege or favour that the State is expected to bestow on persons with disabilities as "beneficiaries" insofar as its economic capacity permits. In other words, the value of accessibility has conventionally not been considered as a part of the mainstream development agenda but rather as a therapeutic and palliative measure to push persons with disabilities towards the attainment of the ableist goal of "normality".

It is important to guard against perceiving disability or disabled persons as problems to be addressed through accessibility. Rather, accessibility must be viewed as an instrument bringing positive transformation and giving impetus to the ability of disabled persons to exercise their freedom of choice. This type of critique helps generate questions about the legal regime in India, which is explained below.

A reflection on which of these two approaches to accessibility resonates in judicial discourse in India is one of our major objectives in this paper. We also seek to explore the role of the right to accessibility in fostering and enhancing the right to mobility of persons with disabilities in public streets as agents, co-determinants and equal citizens of India. We consider the extent to which accessibility, as an instrument of empowerment, fosters parity, rather than charity, and an inclusive, egalitarian, just and fair society.

Engagement with these objectives is germane to understanding whether the right to accessibility, as espoused by the legislature and courts in India, has transformative or ableist overtones. We also assess the degree of visibility enjoyed by persons with disabilities when environments are accessible. In this paper, we also advance the claim that effective enforcement of accessibility, as an enabler right, would have a ripple effect on

the enjoyment and enforcement of other human rights for persons with disabilities. The principle of accessibility, being malleable and adaptive in nature, converges with other human rights (such as the rights to life, freedom of movement, equality and reasonable accommodation). It is interesting to examine the extent to which the judiciary has leveraged this vital quality of accessibility and acted as a catalyst for transformation in the lives of persons with disabilities through its enforcement of the right to accessibility. The same will also help in determining how the enforcement of this right has generated supportive doctrines, standards or principles.

## 3. Bird's-Eye View of the Accessibility Regime in India

This section is divided into three parts to facilitate systematic discussion. Firstly, we briefly outline the constitutional mandate. This is followed by a concise account of the statutory regime dealing with accessibility vis-à-vis persons with disabilities. We conclude this section with a discussion of policies and campaigns initiated by the Union, as well as State Governments.

### 3.1. Constitutional Mandate

The Republic of India has a written constitution that entered into force on 26 January 1950. Because India was a British colony before acquiring independence on 15 August 1947, most of the Constitution's provisions resonate with the Government of India Act 1935. There is, however, one prominent exception, in the form of an entrenched Bill of Rights inhibiting the Legislatures and the Executives from enacting laws or taking any action contrary to or in breach of the guaranteed rights. Being a federal State, there is a systematic allocation of powers between the Union and the provinces. Unlike classical federations, such as the USA and Australia, in India, the allocation of powers between the Union and the provinces is organised at three levels. On certain matters of national importance, the Union has exclusive competence to make laws. On matters that are predominantly local and regional, provinces have exclusive legislative power. Matters relevant to both are allocated concurrently to the Union and the provinces, with primacy being given to the former in the event of its laws conflicting with those of the latter. The Constitution was amended to facilitate the devolution of powers to local and self-governing institutions, such as panchayats, municipalities and municipal corporations. The subjects of transportation, planning and built environment are allocated to all levels of government—national, provincial and local government.

In India, the legal system is predominantly a common law system, but in some pockets of the north-east and scheduled areas, a tribal legal system is also prevalent. The Constitution envisages a parliamentary form of government and a dualist approach to international treaties. However, Article 51 obliges the government to promote international law. When international treaties are incorporated into domestic law through legislation, they acquire a status akin to domestic law and can be enforced in domestic courts, in accordance with the incorporating legislation. Relevant international agreements that have been incorporated into domestic law include the UNCRPD, Universal Declaration of Human Rights, 1948 (UDHR), International Covenant on Civil and Political Rights, 1966 (ICCPR) and International Covenant on Economic Social and Cultural Rights 1966 (ICESCR).

Section 2 (d) of the Protection of Human Rights Act 1993 defines human rights as:

> "... the rights relating to life, liberty, equality and dignity of the individual guaranteed by the Constitution or embodied in the International Covenants and enforceable by courts in India".[1]

---

[1] See also Protection of Human Rights Act 1993, Section 2(f). The same is relevant as it defines "International Covenants" to mean the International Covenant on Civil and Political Rights and the International Covenant on Economic, Social and Cultural Rights adopted by the General Assembly of the United Nations on the 16 December 1996 and such other Covenant or Convention adopted by the General Assembly of the United Nations as the Central Government may, by notification, specify".

There is an Apex court (the Supreme Court) that acts as an arbiter in the event of disputes between the Union and the provinces. As a post-colonial constitutional court, the Supreme Court also has original jurisdiction to interpret the Constitution and vindicate violations of fundamental rights. Generally, in each province, there is a High Court that has a similar original jurisdiction.

The broad and general language of the Bill of Rights gives extraordinary leeway to the judges to unravel its meaning, so as to keep it in step with the changing socio-political and legal landscape. It will be demonstrated below how these courts have elevated the status of persons with disabilities as equal citizens—by problematising the ableism underlying law and focusing on scrupulous enforcement of relevant statutory provisions.

In the absence of any provision in the Constitution prohibiting disability-based discrimination, judges have from time to time performed the role of changemakers and thought leaders, infusing values of empathy, diversity, dignity and accessibility into their interpretation of both the Constitution and the statutes. Lawyers and activists can impress upon courts, during adjudication, the importance of abandoning the medical model of disability in favour of the social model and human rights approach—thereby initiating a shift in focus from the body of an individual to social, economic and political barriers arising out of the design of ableist and inegalitarian society.

Specific statutory duties, like conducting social and accessibility audits, can be effectively enforced by the judiciary as a corollary of the right to life and personal liberty under Article 21 of the Constitution. In this regard, a number of arguments can be advanced, by drawing on the notion of responsive constitutionalism evolved by Professor Rosalind Dixon, to legitimise a more creative and activist role for courts in the domain of disability rights (Dixon 2023b). Notwithstanding the effective functioning of democratic legislative and executive processes, blockages and errors may still occur. This possibility is exacerbated in the case of persons with disabilities. Such people—as a less visible, most marginal, excluded and alienated social group—are likely to find it extremely difficult, if not impossible, to attract adequate attention from legislatures. For instance, a legislature may not be able to foresee the different ways in which laws would impact the rights of persons with disabilities. Such an oversight is evident from the omission in the Indian Constitution of provisions prohibiting disability-based discrimination. Similarly, the Rights of Persons with Disabilities Act 2016 (RPwD Act 2016) condones disability-based discrimination if it is for the attainment of what is considered to be a "legitimate aim", without in any way defining this.

Secondly, legislators may be unable to anticipate the ways in which laws could be efficiently tailored to encompass constitutional norms without being incongruent with the relevant legislative objective. The same is illustrated by the determination of the threshold of disability for the purposes of entitlement to reasonable accommodation or other allied benefits. Similarly, the perfunctory approach of the legislature to affirmative action for persons with disabilities in education also glaringly demonstrates this oversight. Thus, unlike the sphere of employment, where inter-disability reservations are provided, in the education domain, the Parliament has merely provided for a 5% reservation for persons with disabilities without apportioning this amongst people with different types of disability. This asymmetry has not been rationalised. Although accessibility has been recognised as one of the overarching principles underlying the RPwD Act 2016, Parliament has not been able to provide a mechanism with the requisite expertise for its effective implementation and enforcement.

Legislators are likely to have a limited range of life experiences, which may inhibit their imagination about the experiences and needs of certain groups of persons affected by a law. This is self-evident in respect of persons with disabilities because such people seldom attain representation in legislative or executive processes. Their absence from these processes results in a short supply of required lived experience on these bodies.

A legislature may also be vulnerable to the "burden of inertia", thereby justifying an activist role for the courts. Under time and capacity constraints, "the legislators prioritise

issues of greatest concern to the majority, or largest group in society seeking legislative change" (Dixon 2023a) by overlooking issues vital to minorities and oppressed groups.[2] For example, the RPwD Act 2016 was passed virtually on the very last day of the Parliamentary session and without any debate. The Parliament took nine years to incorporate the mandate of UNCRPD as part of the Indian legal order. Despite 75 years of the Indian Constitution, no urgency has been demonstrated by the legislature, either to dispense with condescending language about disabled persons or to do away with stereotypes and myths about their productivity.

The above discussion clearly establishes a justification for interventionist and counter-majoritarian institutions, like constitutional courts. Undoubtedly, this role has been played rigorously and creatively by these courts in India in relation to the protection and enforcement of the rights of persons with disabilities, reinforced by the golden triangle under the Indian Constitution (Articles 14, 19 and 21 in part III). The same is equally legitimised by the explicit obligation on these constitutional courts under Articles 32 and 226. In a way, courts have given an impetus to the evolution of a responsive constitutionalism addressing the democratic dysfunction of the lingering exclusion and alienation of social groups like persons with disabilities (Dixon 2023a; Jain and Jain 2024). Against this background, it is important to reflect on the extent to which the Indian Constitution is cognisant of people with physical and mental disabilities as a social group.

Even a cursory glance over its provisions makes it clear that the Constitution is deeply ableist in its treatment of and reference to disabled people. There is no provision prohibiting disability-based discrimination, nor is there any mandate for the State to initiate affirmative action for persons with disabilities. On the contrary, the Constitution is replete with provisions making "persons with unsound mind" and physical incapacity ineligible for holding or occupying public office and voting and for contesting elections (Jain 2021).[3] The Constitution is ambivalent about recognising persons with disabilities as one of the weaker sections of society. In this light, it is important to explore the trajectory of the right to accessibility in the realm of the Indian Constitution.

Arguably, accessibility is inherent and quintessential to almost all fundamental rights guaranteed under the Indian Constitution. For example, the right to equality before the law and equal protection from the law (guaranteed by Article 14) necessarily covers access to schools and colleges for persons with disabilities so they can engage in education on an equal basis with others. Similarly, the right to move freely throughout the territory of India (guaranteed by Article 19(1) (d)) includes access to public streets and roads for persons with disabilities, so they can navigate freely on an equal basis with others. It also obligates the State to make streets and roads barrier-free. Even if such an obligation were to take the form of a reasonable accommodation duty, not imposing disproportionate or undue burdens, the State would still be duty-bound to take positive measures, including the allocation of resources, to safeguard accessibility for persons with disabilities. Other freedoms, such as the right to freedom of speech and expression, to form associations or to reside and settle in any part of India, or to practise or carry out any profession, are also permeated by the overarching principle of accessibility.

In addition, accessibility also makes the right to life and personal liberty (guaranteed by Article 21 of the Constitution) more meaningful and existentially relevant for disabled persons. In other words, accessibility expands the conception of life and personal liberty beyond mere existence and enables disabled persons to pursue a more holistic state of being. The symbiosis of dignity with accessibility adds further richness to this perspective (Waldron 2023).[4]

---

2 See *United States v. Carolene Products Co.*, 304 U.S. 144 (1938), footnote 4 ("There may be narrower scope for operation of the presumption of constitutionality when legislation appears on its face to be within a specific prohibition of the Constitution. . .").

3 Constitution of India 1950, Articles 102, 191, 317(3)(c), and 326.

4 Articulating the conception of "dignity", Waldron, in his recent book, has identified three manifestations of dignity: dignity as a specific legal right; dignity as the basis of all human rights; and dignity as an intrinsic

This integrated reading of Articles 14, 19 and 21, metaphorically read as the "golden triangle" of the Constitution, provides an enriched and conceptually thick legal architecture for the realisation of the fundamental freedoms and human rights of disabled persons, standing in juxtaposition with the ableist non-discrimination clauses enshrined in Articles 15 and 16. Such an interpretative praxis may also present the doctrine of due process of law in its best moral light (Dworkin 1997), with an emphasis on inclusion and respectful difference. In other words, the convergence of accessibility with the golden triangle—involving rights to equality, facets of freedoms, life and personal liberty—is bound to provide an impetus for the creation of a more egalitarian and inclusive society.

This approach would enable the State to prioritise various obligations pertaining to accessibility with a bearing on different types of rights. The State and courts may also be guided in this pursuit by resource and technological considerations. Furthermore, accessibility, as an overarching principle enshrined in Article 3 of the UNCRPD, may be deployed by courts in India as a tool to create rights-enabling conditions[5] for persons with disabilities in connection with all the fundamental rights guaranteed by Part III of the Indian Constitution.

This discussion shows that, even in the absence of express reference to the rights of disabled persons in the Indian Constitution, a broad interpretation of its provisions by the courts—sensitive to and guided by the letter and spirit of the UNCRPD—can afford more than adequate human rights protection for disabled persons.

*3.2. The Statutory Regime*

Until 1995, there was no specific legislation in India addressing the rights of persons with disabilities. The Parliament enacted the Persons with Disabilities (Equal Opportunities, Protection of Rights and Full Participation) Act, 1995 (PwD Act 1995) as part of the commitment of the Government of India to commemorate the Proclamation on the Full Participation and Equality of People with Disabilities in the Asian and Pacific Region. The PwD Act 1995 was predominantly influenced by the medical model of disability. In 2006, the United Nations adopted a historic specialised convention, the UNCRPD, which was signed and ratified by India in 2007. The convention entered into force in 2008. To incorporate the mandate of this treaty, India replaced the PwD Act 1995 with the RPwD Act 2016. In the interests of providing a full account of the accessibility regime in India, we must compare the relevant provisions of RPwD Act 2016 with those of the UNCRPD and the PwD Act 1995.

The resemblance between the relevant provisions of the RPwD Act 2016 and those of the UNCRPD is striking in that both fail to define accessibility. Instead, both instruments define "universal design" (RPwD Act Section 2(ze) and UNCRPD Article 2 paragraph 5) and, in addition, the former also defines "barrier" (RPwD Act, Section 2(c)). The term "universal design" is defined in broadly similar terms by both instruments. Section 2(ze) of the RPwD Act 2016 reads:

> "universal design" means the design of products, environments, programmes and services to be usable by all people to the greatest extent possible, without the need for adaptation or specialised design and shall apply to assistive devices including advanced technologies for particular groups of persons with disabilities.

Careful consideration of this definition reveals that the legislature has confined the term to "assistive devices including advanced technologies for particular groups of persons with disabilities". Article 2 of the UNCRPD adopts a broader approach, specifying that it "shall not exclude assistive devices for particular groups of persons with disabilities where this is needed". The former adopts a stipulative approach to the subject matter to which

---

characteristic of the right holder underlying the form and structure of rights. All these forms are echoed in the interpretative praxes of the courts in India.

[5] Rights-enabling conditions are relatively flexible and open-ended compared with the capability approach propounded by Martha Nussbaum, wherein the capabilities are specifically enumerated.

universal design applies, whereas the latter, with its non-exhaustive tenor, leaves open a whole host of domains for the application of universal design. From the definition of universal design enshrined in the RPwD Act 2016, it is evident that the legislature does not intend to attain accessibility exclusively through universal design—the limitations of which are recognised. It is supplemented by the notion of "barrier". Section 2(c) of the RPwD Act 2016 reads:

> "barrier" means any factor including communicational, cultural, economic, environmental, institutional, political, social, attitudinal or structural factors which hampers the full and effective participation of persons with disabilities in society.

The definition of "public facilities and services" in Section 2 (x) of the RPwD Act 2016 is also relevant. It defines this phrase as including "all forms of delivery of services to the public at large, including…Transportation".

A close examination of these definitions demonstrates that, for the design of products and services to be usable by all in the domain of transport, environmental and structural factors play an important role. If these factors do not remain sensitive to diversity and merely consider the needs and requirements of the able-bodied, then the policy of the government cannot be said to be in furtherance of neither the letter or the spirit of the UNCRPD or the RPwD Act. To make transportation inclusive and usable by all, policymakers have to eliminate the attitudinal deficit and demonstrate the necessary will power to abandon ableist perspectives. A paradigm shift is also necessary in political processes to advance the agenda of inclusive transportation. There should be willingness to offset any additional costs of accessibility with the benefits of opening public facilities and services to hitherto excluded sections of society, including disabled and elderly persons. This point is particularly relevant, given the Parliamentary recognition of transportation as a public facility. Public spaces should belong to all, irrespective of difference. In other words, the provisions for accessible transportation and the recognition of transport as a public facility in the RPwD Act 2016 are rights-enabling provisions for the effective enforcement thereof, and key to challenging the ableist domination of public space.

The term accessibility does appear in certain provisions of the RPwD Act.[6] Its reference is mostly sectoral, with several substantive provisions of the Act addressing accessibility in particular spheres. Section 8 deals with armed conflict, humanitarian emergencies, and natural disasters, and obliges appropriate authorities, while undertaking reconstruction activities, to remain sensitive to accessibility for persons with disabilities.[7] Section 11 deals with electioneering and emphasises accessibility for persons with disabilities in respect of polling booths and other allied matters. Recognising the interface of the right to education with accessible transport for persons with disabilities, Section 16 (viii) mandates governments—at both the Union and Provincial levels and local and self-government authorities—to provide transportation facilities to children with disabilities, as well as caregivers of any such children with high support needs. Read alongside Article 21 A of the Constitution of India, which guarantees the right to free and compulsory education to all children in the age group of 6 to 14 years, non-compliance with Section 16 (viii) constitutes a violation of this right.

Section 40 imposes obligations on the Union Government, in consultation with the Chief Commissioner of Disability, to formulate accessibility standards for the "physical environment, transportation, information and communications, including appropriate

---

[6] RPwD Act, Section 8: Protection and safety, Sub-section (4); RPwD Act, Section 11: Accessibility in voting; RPwD Act, Section 40: Accessibility; RPwD Act, Section 41: Access to transport, Sub-section (1), clause (a); RPwD Act, Section 44: Mandatory observance of accessibility norms; (RPwD Act, Section 45: Time limit for making existing infrastructure and premises accessible and action for that purpose, Sub-section (2); RPwD Act, Section 46: Time limit for accessibility by service providers; RPwD Act, Section 65: Functions of Central Advisory Board on disability, Sub-section (2), clause (e); RPwD Act, Section 71: Functions of State Advisory Board on disability, Sub-section (2), clause (e); RPwD Act, Section 100: Power of Central Government to make rules, sub-section (2), clause (g). In some sections, the term "accessible" is also used.

[7] It is submitted that this provision sounds more like an obligation than an enforceable right.

technologies and systems, and other facilities and services provided to the public in urban and rural areas".[8]

Section 41 obliges governments, at both Central and Provincial levels, and local and self-government authorities, to provide the following:

(a) facilities for persons with disabilities at bus stops, railway stations and airports conforming to the accessibility standards relating to parking spaces, toilets, ticketing counters and ticketing machines; (b) access to all modes of transport that conform the design standards, including retrofitting old modes of transport, wherever technically feasible and safe for persons with disabilities, economically viable and without entailing major structural changes in design; (c) accessible roads to address mobility necessary for persons with disabilities.

It is also a duty of such governments and authorities to prepare schemes and programs "to promote the personal mobility of persons with disabilities at affordable cost to provide for, —(a) incentives and concessions; (b) retrofitting of vehicles; and (c) personal mobility assistance".

The structure of Section 41 (1) is extremely complex. From one perspective, the duty to take appropriate measures to provide accessible transport may be characterised as structural and anticipatory. From another perspective, in the absence of any provision to tackle non-compliance, and with terminology such as "suitable measures", it gives the impression of being simply aspirational or declaratory. However, the use of the term "shall" complicates the matter and suggests that non-compliance is a breach of public function warranting the issuance of writs and directions from the constitutional court under Articles 32 and 226 of the Constitution of India. On this view, from the standpoint of analytical jurisprudence, this duty may be characterised as the correlative of a right to accessibility for persons with disabilities—although that right is not explicitly conferred by this provision. The right–duty correlation could then be enforced in both civil and constitutional courts, depending upon the nature and gravity of the non-compliance. Section 89 of the RPwD Act 2016 provides "[a]ny person who contravenes any of the provisions of this Act, or of any rule made thereunder... shall be punishable with fine...". This section has a very narrow compass in that it provides a redress against violations of the provisions of the RPwD Act 2016 by "any person", but it does not account for the violation of structural and systemic duties by various specific authorities, nor the State more generally.

Although the mandate of Sections 84 and 85 is to specify, for each district, a Court of Session to be a special court to try the offences under RPwD Act 2016 and appoint a Public Prosecutor (an advocate, who has been in practice as an advocate for not less than seven years), for the purpose of conducting cases in such courts, compliance with both these provisions is very low, with an almost total absence of such courts or prosecutors in most districts.

The RPwD Act 2016 lacks effective mechanisms for tackling breaches of structural duties.

The Ministry of Road Transport is mandated to play a pivotal role in attaining the objectives underlying the provisions at both Central and Provincial levels. Section 40 imposes a duty on the Union Government to develop rules and accessibility standards. The Government of India has, however, not appointed a Chief Commissioner of Disability for at least the last five years, and Section 40 has therefore not yet been implemented. Section 45 had originally prescribed a timeline of 5 years for the same, but it has been diluted by repeated extensions. The tone of Section 46, which imposes obligations on service providers to adhere to accessibility standards, is no different. Thus, in light of the watered-down content of Sections 45 and 46, the mandatory obligation of Section 44 on governments, to observe accessibility norms, sounds hollow.

In this connection, reference may also be made to Rule 15 of the Rights of Persons with Disabilities Rules, 2017, as amended in 2023 (Rights of Persons with Disabilities

---

8    RPwD Act 2016, sections 6(2)(e) and 71(2)(e) laying down the functions of Central and State Advisory Board, and section 40 (g) empowering the Central Government to frame Rules.

(Amendment) Rules 2023), which obligates compliance with certain standards relating to physical environment, transport and information and communication technology by every establishment during the construction of public buildings, as specified in the Harmonised Guidelines and Standards for Universal Accessibility in India (2021), issued by the Ministry of Housing and Urban Affairs, Government of India.[9]

A comparison of the accessibility regime established by the RPwD Act 2016 with the transportation and roads provisions of the former PwD Act 1995 helps clarify whether the former has made any noteworthy advance in harnessing the principle of accessibility, in light of India's ratification of the UNCRPD. Under the RPwD Act 2016, the heading of Section 41 is "Access to transport". On the other hand, the headings of the sections under the PwD Act 1995 are 'Non-Discrimination on the Road and Non-Discrimination in Transport'. It is striking that, unlike the PwD Act 1995 (which imposed an obligation on both the Central and Provincial governments and local and self-government authorities), the heading under the former is couched in neither duty-imposing nor in rights-conferring language.[10] Another important difference between the two enactments is that the RPwD Act 2016, while obligating all forms of government to take suitable measures to provide accessible transport and roads for people with disabilities, is not hedged around with any conditionalities, such as those based on economic capacity and development, whereas the obligation under the PwD Act 1995 was. It is, therefore, possible to argue that the nature of the duty under the RPwD Act 2016, regarding accessible transportation and roads, can be construed as categorical. Of course, this observation is qualified by and subject to the analysis above about the nature of this obligation. Finally, in terms of details, the PwD Act 1995 was more articulate regarding the measures to be adopted for accessibility. On the other hand, the RPwD Act 2016 imposes a substantive obligation on the Central Government to lay down accessibility standards in consultation with the Chief Commissioner of Disability, through rules to be made under the Act. Although this is appropriate, the mechanism created for discharging the obligation appears to be half-hearted, as it is assumed that the Chief Commissioner for Persons with Disabilities has the institutional expertise to suggest and recommend the accessibility standards. In our opinion, given that the subject matter of accessibility is a specialised area of knowledge, it would have been more appropriate had the Parliament envisaged an accessibility board consisting of access auditors and allied experts along the lines of the US Access Board. Thus, overall, a comparison between the two Acts demonstrates that although, in some respects, the RPwD Act 2016 is stronger than the PwD Act 1995 regarding accessibility obligations, it cannot be argued that the ratification of the UNCRPD has provided any significant impetus for strengthening or harnessing it.

It is submitted that, although there has recently been some excitement in the Union Government's Department of Empowerment of Persons with Disabilities, in connection with the evolution of accessibility standards, the absence of an operating Chief Commissioner of Disability would make the agency less bold or creative in its implementation. The Secretary, who has the additional role of Chief Commissioner of Disability, is currently engaged in this process, but his approach is top-down in that his consultations are largely confined to experts and stakeholders based in Delhi.

*3.3. Policies, Guidelines and Campaigns*

In this section, we briefly analyse relevant government initiatives in the form of policies, guidelines and campaigns.

---

[9]   See Notification number O-17/4/2022-works-3-UD, dated 18 October 2022, as amended from time to time.

[10]   We emphasise headings because in India, while interpreting statutes, courts do rely on section headings to resolve ambiguities.

3.3.1. Accessible India Campaign

In order to implement goal 3 of the Incheon Strategy on the Asia Pacific Decade for Persons with Disabilities: 2013–2022, the Department of Empowerment of Persons with Disabilities (DEPwD) introduced the Accessible India Campaign in 2015 (India.gov.in National Portal of India n.d.). This was a nationwide campaign for achieving universal accessibility for persons with disabilities (Incheon Strategy to "Make the Right Real" for Persons with Disabilities in Asia and the Pacific 2012).

One of its objectives was to enhance the proportion of accessible government buildings. It included the following targets: to conduct accessibility audits of at least fifty and twenty-five of the most important government buildings in tier one and tier two cities, respectively, and to convert them into fully accessible buildings by July 2016 (Accessible India Campaign). By 30 September 2022, 1100 Central Government buildings had been audited and made accessible. By contrast, of 4522 identified and audited State Government buildings, only 967 had been made accessible (with the installation of accessibility features such as ramps, lifts, toilets and parking) (Standing Committee on Social Justice and Empowerment 2023, p. 43). The report also furnishes data on the accessibility of air, rail and bus transport. In these sectors, the government is also well behind the target. To date, there have been four extensions: a first extension, from 2016 to 2018; a second extension, from 2018 to 2020; a third extension, from 2020 to 2022 (Nath 2022; Sharma 2023); and a fourth extension, from 2022 to 2024 (Standing Committee on Social Justice and Empowerment 2023, pp. 42–43).

The limitation of this campaign is its pivot towards urban areas and selected government buildings, to the complete exclusion of rural areas, where the majority of people with disabilities live. Even in respect of urban areas, there is a want of attention to public streets and roads and non-governmental buildings like theatres, restaurants and other places of recreation.

It is interesting to note how the objectives pertaining to the built environment underlying this campaign were articulated:

> The objective is to increase accessibility in government buildings. Provision of features of accessibility such as staircases, ramps, double height handrails, tactile paths in corridors, wide entry gates, reserved parking and disabled friendly toilets, accessible elevators, etc. to be made. (Accessible India Campaign: An Inclusive Society Creates a Sashakt Bharat 2015, p. 7)

Close attention to this wording suggests the adoption of a perfunctory and blinkered approach, in that the focus is on increasing accessibility to specific elements of the environment, rather than fostering a more holistic approach to achieving a barrier-free environment. This approach, apart from being narrow, is not inclusive and comprehensive, as the concentration is merely on certain selective accessibility initiatives, such as the installation of ramps, disabled-friendly toilets, accessible elevators, etc. Clearly, there is hardly any focus on the physical environment in its entirety, including streetscapes and the public realm. For example, there is no mention of enhancing accessibility to roads or making public streets accessible and friendly for persons with disabilities. As a matter of fact, without the recognition and enforcement of street accessibility in public realm, provisions for accessible buildings are asymmetrical.

The narrowness of this focus stands in contrast with the definition of barrier in the RPwD Act 2016, in which the physical environment is only one factor. Here, however, there is a complete lack of attention to other elements affecting accessibility, such as communicational, cultural, institutional, political, social and attitudinal factors. This reflects a top-down approach by the State, rather than a bottom-up approach in which people with disabilities are fully involved in both the identification and the elimination of barriers. This exclusive focus on the physical environment means the campaign loses sight of other important aspects of accessibility and does not address crucial issues, such as inaccessible communication, or the prevalence of negative stereotypical attitudes. Without

the proper sensitisation of officials to the potential of disabled persons and their rights, accessibility to the physical environment would not mean anything but tokenism.

In this light, we are constrained to observe that the government is perceiving people with disabilities as objects worthy of attention rather than as active and political subjects with aspirations and desires.

### 3.3.2. Draft National Policy for Persons with Disabilities

It is not clear why the Union government has not been able to roll out the final version of the policy for disabled persons by revising the now dated version from 2006. At any rate, the draft policy in respect of accessibility does not present a clear government stand on accessibility. There is no indication in the policy expressly recognising accessibility as a substantive right of disabled persons. There are, instead, ambiguous statements such as the following: "[w]hen we consider the disabled persons, accessibility draws significance as it is the primary tool for their empowerment and inclusion" (Public Notice Inviting Comments on the draft National Policy for Persons with Disabilities 2021, para. 9.1). Furthermore, the principle of accessibility contained in Article 9 of UNCRPD "mandates the Member States to take measures to eliminate obstacles and barriers to accessibility. . ." (Public Notice Inviting Comments on the draft National Policy for Persons with Disabilities 2021, para. 9.4). It thus demonstrates the ambiguity in the policy about the exact status of accessibility as either a right or a correlative duty. Prima facie, it is seen as a stand-alone obligation for the State. Moreover, the observation in the policy that the Accessible India Campaign fosters the social model of disability is deeply problematic because, while identifying barriers and restricting its focus to the physical environment, the campaign appears to be more concerned with bodily impairments rather than socio-economic and political barriers.

### 3.3.3. Harmonised Guidelines and Standards for Universal Accessibility in India (2021)

Unlike the draft policy, "the Harmonised guidelines have been envisioned with a key guiding philosophy of universal design contextualised towards Indian perspectives of built and socio-cultural environments" (Harmonised Guidelines and Standards for Universal Accessibility in India 2021, p. 12). The objective underlying these guidelines is to transform the approaches, attitudes and deliverables of the State and civil society towards built environment conceptualisation, retrofitting and implementation. There appears to be a clear shift from a "minimum possible adaptation" (UNCRPD Article 4(1) (f)) and "minimum standards and guidelines" (UNCRPD Article 9(2)(a), read with Article 9 (2) (h)), envisaged in the UNCRPD) to ". . . an approach for creating best practices of universal design and inclusion. . ." (Harmonised Guidelines and Standards for Universal Accessibility in India 2021, p. 12). Enhanced sensitivity and responsiveness towards the needs of diverse users, including persons with disabilities, women, older people, children and several other vulnerable sections of society is at the fulcrum of these guidelines. Five important features underlying them are as follows: human centricity, universal accessibility, equity and inclusion, safety, and inclusive participation (Harmonised Guidelines and Standards for Universal Accessibility in India 2021, p. 12).

In light of this paradigmatic shift, the existing guidelines need to be revised if the following objectives are to be achieved. The first is bringing a shift from a barrier-free environment to a universal design approach; the second is enhancing readability so as to facilitate implementation (Harmonised Guidelines and Standards for Universal Accessibility in India 2021, p. 12).

This paradigm shift must be translated to ". . .the built environment by enabling wider choices in access, incorporation of technologies, affordable and customizable designs, developing innovative alternatives and most importantly by educating built environment professionals and users towards a holistic approach to accessibility" (Harmonised Guidelines and Standards for Universal Accessibility in India 2021, p. 12).

One of the most progressive aspects of this guideline is the adoption of a humanistic approach to the notion of accessibility. "Accessibility is a multi-layered, multi-dimensional

and multi contextual aspect of the built environment. Amidst various existing definitions of the term, these guidelines specify the key framework for accessibility into three key dimensions i.e., Information, Infrastructure and Services/Management" (Harmonised Guidelines and Standards for Universal Accessibility in India 2021, p. 17).

Thus, to a large extent, this approach represents a move away from the individualised and medicalised notion of disability. With the adoption of this approach, persons with disabilities are no longer viewed as mere passive recipients of privileges; rather, the State now recognises them as co-participants, in the processes of nurturing the value of accessibility, on an equal basis with others. If these guidelines are implemented with rigour, they have the potential to bring about a much-needed transformation in the lives of persons with disabilities. The main advantage of "a design-for-all" perspective is to decentre physical and mental disability from the discourse of accessibility and to enlarge its parameters so as to extend it to other social groups, as well as persons with disabilities.

## 4. Judicial Enforcement of Accessibility

In this section, we will briefly analyse the landmark judicial pronouncements in India relating to the accessibility of the built environment. To reiterate, despite the transition from the PwD Act 1995 to the RPwD Act 2016, alongside India's categorical ratification of the UNCRPD, there has been no qualitative transformation in the protection and enforcement of accessibility. Moreover, as will be demonstrated below, although many cases were filed during the regime of the PwD Act 1995, the litigation has carried over into the era of the RPwD Act 2016. Therefore, a strict chronological analysis may not be helpful for drawing additional insights, and we will instead analyse the judgements according to the different types of contexts in which the cases arose. Although the focus of this paper is on street accessibility, an evaluation of the judgements in its other domains is helpful in deducing broad principles which may have relevance for streets.

### 4.1. Street and Environmental Accessibility

An obvious starting point is court decisions under the PwD Act 1995. In *Integrated Disabled Employees Association v State of West Bengal* (2008), the petitioners sought directions from the Calcutta High Court to State authorities concerning the implementation of Section 44 (Non-Discrimination in Transport) and Section 45 (Non-Discrimination on Roads) of the PwD Act 1995. Finding for the petitioners, the Court directed the authorities to implement Section 45 by installing auditory signals and tactile surfaces at pedestrian crossings, for the benefit of persons with visual impairments, as well as curb cuts and slopes on pavements to facilitate access for wheelchair users. The Court also directed that warning signals should be installed at appropriate places for such persons.

The petitioners had prayed for implementation of the provisions of the PwD Act 1995, as well as of the UNCRPD's underlying principles, in order to ensure a disabled-friendly and accessible public transport system. The petitioners further contended that the failure of the State Government to implement the provisions of the PwD Act 1995, even though thirteen years had passed since its commencement, amounted to a violation of Articles 14, 19 and 21 of the Constitution of India. Unfortunately, the court did not address this significant contention and confined its attention to low-hanging fruit, such as reserving a few seats for blind persons on buses. In respect of a Memorandum issued by the Government regarding the conferral of certain benefits upon persons with disabilities, the court merely made an off the cuff observation about the need for scrupulous adherence to the Memorandum. The case proved to be a missed opportunity to constitutionalise—under articles 14, 19 and 21—the accessibility mandate under Sections 44 and 45 of the PwD Act 1995. The court also missed the opportunity to draw on the conceptual reorientation introduced by UNCRPD in its approach to accessibility.

Similarly, *Rajive Raturi v Union of India* (2018)[11] was concerned with the issue of safe accessibility for persons with visual disabilities of pavements and to roads and other transport. The petitioners also sought the provision of proper and adequate access to public places. In this judgement, Justice A K Sikri observed:

> The vitality of the issue of accessibility vis-à-vis visually disabled person's right to life can be gauged clearly by the Supreme Court's judgement in *State of Himachal Pradesh v Umed Ram Sharma* (1986) where the right to life under Article 21 has been held broad enough to incorporate the right to accessibility. (*Rajive Raturi v Union of India* 2018, para. 12)

The Madras High Court followed suit in *Vaishnavi Jayakumar v The State of Tamil Nadu* (2022). In this case, a Public Interest Litigation action (PIL) was filed to challenge a tender, issued by the Road Transport Department, for procuring 1107 high-floor buses—on the grounds that this was not consonant with the law laid down in *Rajive Raturi*. Although the government accepted the petitioner's standpoint, the State disguised design barriers as "practical difficulties such as road conditions, inundation during rainy seasons, the longer length of low floor buses, higher cost, lack of competition (in procurement inasmuch as only two manufacturers alone manufacture and supply low floor buses), maneuvering space in the roads; lack of proper platforms in bus stops, enabling them to get-in (sic) etc.". (*Vaishnavi Jayakumar v The State of Tamil Nadu* 2022, para. 4).

To test whether these arguments held any water, the court directed the State to run one low-floor bus on specified routes (including difficult ones with narrow roads and sharp corners) with relevant stakeholders (including disabled people) on board. Based on the lived experiences of these stakeholders, the court debunked the ableist stance of the State and cautioned it against perpetuating the status quo by emphatically asserting that "[t]here may be some difficulties... But over all... the low floor buses are very much pliable and all the other factors cannot be put against but only need to be improved" (*Vaishnavi Jayakumar v The State of Tamil Nadu* (2022), para. 9). Digging deeper into the root cause, the court directed that "the respective Municipal Corporations shall strive to continuously improve the quality of the roads, their maneuvering capacity and shall scientifically lay the bumps/speed breakers enabling the smooth running of these low floor buses".[12]

In sharp contrast is *Integrated Disabled Employees Association v State of West Bengal* (2008), which also involved the State obligation to enhance accessibility in the transport sector. The judgement in Vaishnavi Jayakumar is an advancement on the Integrated Disabled Employees Association (2008), where the court merely focused on reserving seats on buses and overlooked the structural and systemic aspects of accessibility obligations.

Just as street inaccessibility impedes pedestrian mobility, vehicular inaccessibility obstructs transport-based mobility; therefore, cases dealing with the latter also have a bearing on the former. After all, the underlying objective is to adopt measures facilitating freedom of movement—a fundamental right guaranteed by Article 19(1) (d) of the Constitution of India.

In a similar vein, the Bombay High Court, in *High Court of Judicature at Bombay on its own Motion v Municipal Corporation for Greater Mumbai* (2023), recognised the grievances of disabled persons regarding inaccessible footpaths in the city of Mumbai. The court took cognisance of an email sent by a disabled person to an advocate, highlighting the inaccessibility caused by the installation of poles/bollards at the entrances of footpaths in Mumbai. Appointing this advocate as the Amicus in the matter, the Court issued

---

11  *Rajive Raturi v Union of India*, (2018) 1 SCC 413. Although the petition was broadly vindicating the right to accessibility of visually challenged people, the court issued a number of general directions to inform the built-in architecture of the country in light of the principle of accessibility. See also *Disabled Rights Group v Union of India*, (2018) 2 SCC 397.

12  Ibid. at para. 13. Space constraint prevents us from providing a full-throated analysis of the creative reasoning of this case. What appears to be an extraordinary course for the State is perceived as ordinary by the court, with the recognition of the right to navigation and mobility of persons with disability on an equal basis with others. The Court is clearly advocating for a design-for-all approach.

notices to the Municipal Corporation for Greater Mumbai and the State of Maharashtra to proceed further in the matter. Implicit in this short but important order is the fact of a court categorically recognising the accessibility interest of disabled persons as part of the public interest.

Reference may be made to yet another PIL filed by an advocate seeking the recognition of accessibility as inherent to the golden triangle of Articles 14, 19 and 21 of the Constitution of India (*Pankaj Mehta v Union of India* (2022)). The advocate sought to read relevant provisions of the RPwD Act 2016 (Section 44) conjointly with the aforementioned provisions. The petitioner sought appropriate directions from the court with a view to ensuring that lifts, footpaths on bridges and other public amenities and structures were convenient and fully accessible for "specially-abled" and senior citizens of India. Upon the notice of the High Court, the Government of Delhi, unlike that of Madras in the previous case, informed the court that it would remedy the situation by taking appropriate action in a timely manner, instead of adopting a denial or adversarial stance.

Reference may also be made to *Jyoti Singh v Nand Kishore* (2021), an ongoing litigation in Delhi High Court in respect of measures to be taken by the Delhi Government to secure an enabling environment for persons with disabilities. The case reached the High Court by way of an appeal from the Motor Accidents Claim Tribunal. It concerned an accident involving a young schoolgirl, resulting in her losing the ability to walk for the rest of her life. The case assumed significance because, while disposing of the appeal regarding the level of compensation payable, the court addressed the contention of the petitioner that "concerted efforts need to be made to ease movement of wheelchair-bound persons" (*Jyoti Singh v Nand Kishore* (2021), para. 3). It is arguable that judicial minimalism must restrain the court from entertaining such a general contention in the absence of a claim that any legal right has been violated. However, the court was swayed by the future prospects of the petitioner and decided to expand the scope of the litigation to recognise and enforce the interest of accessibility as part of public law. In other words, in the instant case, the court recognised the locus standi of the petitioner to raise the above contention by implicitly identifying the matters of accessibility and enabling environment as having a "public interest element" (*Samridhi Devi v Union of India* (2005), following the aforementioned ruling of the Bombay High Court in *High Court of Judicature at Bombay on its own Motion v Municipal Corporation for Greater Mumbai* (2023)).

Subsequently, the court could have issued appropriate directions and disposed of the matter. Instead, the Court decided to assume a somewhat executive role in ensuring the initiation of the measures towards achieving the end of having an enabling environment across the city. For this purpose, the court issued a number of programmatic directives to the Government of Delhi. In a collaborative spirit, the court was informed that "this exercise would be best undertaken under Sections 2(x), 2(zd), 2(ze), 28, 40, 41, 42, of the Rights of Persons with Disability Act, 2016" (*Jyoti Singh v Nand Kishore* Order 16.12.2021, para. 2). Accordingly, the court issued the appropriate direction for the appointment of the Nodal Officer to conduct a Social Disability Audit in a time-bound manner. The Court expressed the hope that "...in three months streets not less than two kilometres each in the South, East, North, West and Central regions will be identified, and made ready, in terms of the social disability audit...". (*Jyoti Singh v Nand Kishore* Order 16.12.2021, para. 10). Constitutionalising these directions, the court observed:

> The non-availability of requisite and enabling infrastructure for persons with disability is glaring and apparent throughout the city. It is also a violation of the Article 21 of the Constitution of India. Freedom of movement has to be honoured and assured in every way possible, it cannot be restrained by lack of civic amenities.[13]

---

[13]   Ibid. at para. 8. Since the issuance of this order, the matter has continued. Thus, the court was informed during the proceedings in 2022 about the modalities of the "social disability audit" to be conducted by the government. The government informed the court that it would consult the domain experts and the stakeholders during the audit.

Obviously, while the Government has not been able to meet the timelines, the progress of the litigation clearly demonstrates sincerity in its efforts. Further, considering that some accessibility obligations are subject to progressive realisation, there is adequate scope for allowing some space to the government to undertake the required actions. This is the only case in which any court in India has issued the direction to conduct a social disability audit under Section 48 of the RPwD Act 2016. Unfortunately, the Act does not define the term "social audit"; therefore, Section 48 is merely indicative.

Social audit, which is also known as performance audit, refers to a legally mandated tool and process, whereby potential and existing beneficiaries evaluate the implementation of a programme by comparing official records with on-the-ground realities. During a social audit, the beneficiaries and implementing agency come together to discuss and analyse the information and share publicly, on a participatory platform, the report concerning the implementation and progress of a particular programme/scheme. The process of a social audit is not a fault-finding exercise; it is, rather, a fact-finding process (Report No. 8 of 2016—Union Civil Mahatma Gandhi National Rural Employment Guarantee—Social Audit 2016).

We wonder whether a social disability audit is an efficacious means by which to evaluate levels of accessibility. Instead, we suggest that it would have been preferable if the court had ordered an access audit. Although the term "access audit" is neither deployed nor defined in RPwD Act 2016. It is "an assessment of a building, an environment or a service against best-practice standards to benchmark its accessibility to disabled people" (Evans Jones n.d.) and stems from accessibility obligations under the UNCRPD. A social disability audit is appropriate for the evaluation of ongoing schemes and progress; on the other hand, since there is an almost total lack of any activities pertaining to accessibility measures for persons with disabilities, a social disability audit may have little impact. Furthermore, the approach of the learned judge in conceptualising disability, while adjudicating the compensation for the petitioner, is also not free from controversy. He opined the following:

> She would suffer social and personal embarrassment because of her uncontrolled bowel movement. In these circumstances, a just and fair compensation is to be awarded so that she is put in a position as close as to what she could have been without the injury. (*Jyoti Singh v Nand Kishore* Order 11.04.2023, para. 7)

In other words, the learned judge was inclined to compensate the lack of normality which resulted from the injuries due to "tort of Nonfeasance". The opinion of the learned judge was clearly influenced by a negative ontology of disability. It was possible for the learned judge to invoke the positive ontology of disability, and he could have identified rehabilitation of the petitioner post-disability as one of the heads of the compensation. Thus, paradoxically, on one hand, the learned judge perpetuated the medical model of disability but, on the other hand, while pressing for the accessibility of the physical structure of the city, he was somewhat influenced by the social model of disability.

In this connection, reference may also be made to cases dealing with the legal and physical injuries caused to persons with disabilities, due to negligence. However, as will be shown by the following discussion, the law has not yet evolved to imbibe the spirit of the UNCRPD. Let us analyse *Tamil Nadu State Transport Corporation Ltd.*, *Villupuram v Papa* (2016) to demonstrate this point. The case involved a road accident experienced by a blind person, as a result of his collision with a State-owned bus. This resulted in his death. While he was standing on the road, a bus belonging to the Provincial Transport Corporation, driven by its driver in a rash and negligent manner, dashed him, and as a result, he fell on the road, sustained grievous injuries all over his body and died while he was on the way to the hospital. The legal representatives of the deceased filed the petition, claiming a sum of INR 500,000/- as compensation before the Motor Accident Claim Tribunal under Motor Vehicle Act, 1988. After perusing the evidence, the Tribunal awarded INR 376,400/- with interest at 7.5% per annum as compensation. On appeal, the corporation argued that, while attempting to board the bus, the deceased lost his balance because other passengers, who were standing outside the bus, had also attempted to enter and, consequently, he fell in front of the front tyre. It was further argued that the Tribunal, without looking into this

aspect, had wrongly ascribed negligence only to the driver of the bus belonging to the transport corporation. Rejecting the aforementioned argument in the absence of evidence, the High Court of Madras upheld the decision of the Tribunal on the level of compensation. This case exemplifies the lack of action and apathy on the part of both the Corporation and the High Court, because both institutions looked at the incident as a discrete event, rather than looking at it from the broader perspective of inaccessibility.

The notion of negligence and duty of care in order to cover the harms caused to persons with disabilities or elderly people has to move beyond ableist considerations. Therefore, a comprehensive manual on accessibility and a formulation of the policy based on the same is required.

*4.2. Accessibility in Higher Education Institutions*

The Supreme Court expanded the right to accessibility by recognising it to be inherent to the interest of disabled people in pursuing higher education in colleges and universities. In *Disabled Rights Group v Union of India* (2018), the Supreme Court of India grappled with the obligation of Higher Education Institutions (HEIs) to make the built environment accessible. It directed the Union Government, under Section 40 of the RPwD Act 2016, to lay down accessibility standards for colleges and other HEIs within six months of the date of the judgement. In addition, the relevant government was directed to implement the Rights of Persons with Disabilities Rules 2017 within two years, in accordance with Section 46 of the RPwD Act 2016. The Union Government was also directed to create an audit template for HEIs, and relevant governments were required to conduct audits of all HEIs within six months. All HEIs were directed to make themselves accessible in accordance with the RPwD Act 2016 and the rules issued under it within two years, with each institution establishing an enabling unit for persons with disabilities with the role of monitoring the implementation of accessibility standards. As a follow-up to this decision, the Court directed the University Grant Commission India (UGC) to ensure that the Accessibility Guidelines and Standards for Higher Educational Institutions and Universities were finalised, and to commence the inspection of educational institutions and supervise the proactive implementation of the RPwD Act 2016. The UGC informed the Court that it had received responses from a number of HEIs and that the Committee, established by the UGC to prepare accessibility standards relating to infrastructure, pedagogy and curriculum for educational institutions (in consultation with the Chief Commissioner), had initiated work (*Disabled Right Group v Union of India* (2022)).

Similarly, the Bombay High Court, in *Akanksha Vardhaman Kale v Union of India* (2018), converted a petition filed by a petitioner who was a wheelchair-user, into a Public Interest Litigation (PIL) action to challenge inaccessible HEI campuses in the State of Maharashtra. The High Court insisted, among other aspects, on adherence to Sections 40 and 44 of the RPwD Act 2016 and the National Building Code. Issuing a series of directions, the Court observed:

> The State and the Director of Education shall issue instructions to all the Universities to call for compliance report from the educational institutions. In case any completion certificates in respect of the structures, institutions, establishments/buildings as required by the Act are pending with the authorities, then the necessary compliance be sought before issuing the occupancy or completion certificates. In case the occupancy or completion certificates of such buildings were already granted, then such authorities would take necessary steps for creating basic facilities like ramp, washrooms, barrier free access for handicapped persons in such buildings. (*Akanksha Vardhaman Kale v Union of India* 2018, para. 3)

Although, on face value, this is a salutary judgement, it offers little in terms of structural progress. The use of the word "handicapped" shows the lack of disability sensitivity of the judges involved. Neither does the judgement go far enough in imposing sanctions for non-compliance with its directions, nor does the State Commissioner for Disability or

Provincial Government appear to have collaborated in the matter by providing incentives to educational institutions and universities complying with the court directions.

Recently, a petition was filed in the Delhi High Court by Himanshu Goswami against the University of Delhi. The petition challenges a range of issues, including the difficulties faced by visually impaired persons in navigating pavements in the North Campus area. Interestingly, the court made no reference to Sections 40 or 44 of the RPwD Act 2016. Instead, it referred to the "Guidelines for Pedestrian Facilities" published by the Indian Roads Congress in May 2012, and directed the University authorities and the Delhi Municipal Corporation to adhere to the same. In very pithy observations, the court exposed the inadequacy of the authorities' approach:

> We also find that the authorities are shifting responsibility from one to another. While some pavements are claimed to be under the jurisdiction of Municipal Corporation, the responsibility of others is shifted to the Public Works Department. (*Himanshu Goswami v University of Delhi* (2017), para. 6)

Despite having exposed the cavalier approach of the respondent, the court chose to rely on them—requiring them to file status reports, detailing the steps taken to make buildings/installations accessible and produce action plans setting out timelines for the completion of such steps.

To understand the realities on the ground, the court directed that the two campuses, i.e., the North Delhi Campus and South Delhi Campus of the University of Delhi, should be treated as pilot projects. It directed the Commissioners of the North Delhi Municipal Corporation and South Delhi Municipal Corporation, as well as the Engineers-in-Chief of the Public Works Department, to initiate a joint physical inspection of all the pavements around these two campuses and to submit an action plan to ensure that they were made accessible. It further directed steps to be taken forthwith for the removal of all obstructions rendering pavements inaccessible. The court reminded the authorities of their obligations by observing:

> These authorities ought to examine the guidelines suggested by the Indian Roads Congress which note the requirement of the informal commercial activities on the footpaths and the need of integrating it with the design of the footpath facility. If this was properly done the disabled pedestrian would have no difficulty. It is the unstructured, unplanned and hap hazard obstructions which are causing the difficulties. (*Himanshu Goswami v University of Delhi* (2017), para. 10)

> It is made clear that tactile pavement markings shall be continuous, so that there is safe passage for visually impaired persons. (*Himanshu Goswami v University of Delhi* (2017), para. 12)

Close consideration of this judgement shows that the court deposited trust in physical inspection rather than directing the authorities to undertake an access audit. Furthermore, there appears to be a tendency amongst judges to deal with accessibility as a general issue, rather than treating it as a specialist issue requiring expertise. To put it briefly, the courts have undermined the role of access auditors.

*4.3. Accessible Built Environment*

In *Ahmad M. Abdi v State of Maharashtra* (2019), the Court addressed the issue of accessibility for disabled persons in relation to the built environment and issued a number of directions. It observed:

> . . .all our court complexes are conducive and friendly for the differently-abled and towards this end, the Court complexes must have certain features for the benefit of the vulnerable persons such as persons with disability or visually impaired persons. We have to move from disabled friendly buildings to workable and implementable differently-abled friendly court infrastructure. Ramps for such categories of persons must be operable, feasible, tried and tested. Such ramps

should definitely have steel railings and handles. The court infrastructure must also keep in view the accessibility for visually impaired persons and, therefore, court complexes must have tactile pavements and signage in braille for the benefit of visually impaired citizens. (*Ahmad M. Abdi v State of Maharashtra* (2019), para. 12)

While these words sound very positive, the court did not back them up by referring to either relevant enactments or international human rights instruments. In the absence of a proper consultation with relevant experts, the court confined its ruling on accessibility to only one type of disability, i.e., visual disability. If reference to separate toilets for "physically handicapped persons" is broadly construed, people with mobility disabilities may also be covered, but other disabilities were completely overlooked.

In another significant instance, the Gauhati High Court delivered a landmark judgement in *Arman Ali v Union of India* (2019). In this case, an NGO, "Shishu Sarothi", advocated for the rights of a person with cerebral palsy to access a private gym. The petitioner complained of discrimination on the ground of disability by the gym authorities. The gym authorities tried to argue that being a private establishment, they were beyond the purview of Article 12 of the Constitution of India and, thereby, beyond the writ jurisdiction of the High Court. However, the High Court rejected this argument by construing the phrases "public building and public facilities" broadly to extend to private establishments, like gyms. The court emphasised the importance of raising awareness amongst officials of the Province of Assam, and directed the following:

Commissioner and Secretary of the Social Welfare Department, Government of Assam to issue general circulars to all Government and private establishments highlighting the salient features of the 2016 Act and to ensure that public buildings and public facilities and services are accessible by persons with disabilities. Such directions or guidelines may be issued within a period of 2 (two) months from the date of receipt of a certified copy of this order. (*Arman Ali v Union of India* (2019), para. 24)

Exceptionally, the Gauhati High Court also decided to add a financial sanction to its judgement by ruling the following:

Before closing the litigation, Court is of the view that some cost may be imposed both on the State as well as on respondent No. 6. Respondent No. 6 though a private entity has also a duty to ensure that its facilities are friendly to the differently abled. Accordingly, both respondent Nos. 3 and 6 are directed to pay Rs. 50,000 each to the Shishu Sarothi which will be used for the benefit of the specially-abled children who are attending classes in the said centre. Let the deposit of Rs. 50,000 each be made by respondent No. 3 and respondent No. 6 within a period of 2 months from the date of receipt of a certified copy of this order. (*Arman Ali v Union of India* (2019), para. 25)

It is vital to note the change in the paradigm of the court. Unlike the relief-centric approach followed in most of the earlier cases, the focus of the court in these cases was on seeking a transformation in structural design in light of the principle of accessibility. To recognise access to higher education for persons with disabilities on an equal basis with others, the court is able to synergize the right to equality under Article 14 of the Indian Constitution with the right to accessibility. Clearly, this line of reasoning is supported by the dicta of the Supreme Court in *State of Himachal Pradesh v Umed Ram Sharma* (1986), in which the court synthesised the right to life and personal liberty (under Article 21 of the Indian Constitution) with accessibility, by recognising the latter as one of the manifestations of the former.

Similarly, upon careful reflection on the judgement of the Madras High Court, in respect of the accessibility of roads and public transport, it is possible to find synergy between the right to move freely throughout the territory of India and accessibility, because for people with disabilities, the latter is essential for the effective enjoyment of the former

(*Vaishnavi Jayakumar v The State of Tamil Nadu* 2022). In India, the Supreme Court has recognised Articles 14, 19 and 21 as the golden triangle of the Constitution, observing:

> Three Articles of our Constitution, and only three, stand between the heaven of freedom into which Tagore wanted his country to awake and the abyss of unrestrained power. They are Articles 14, 19 and 21. (*Minerva Mills v UOI* (1980), para. 18)

In this light, it is arguable that the right to accessibility is deeply rooted and entrenched by courts, not only through the words used by judges, but also through the subjection of the powers of the State to judicial rulings.

In Section 3, above, we highlighted the significance of the duty to provide reasonable accommodation as a catalyst for the enforcement of accessibility. In this light, a brief reference may be made to the landmark judgement of the Indian Supreme Court in *Vikash Kumar v Union Public Service Commission* (2021), in which Justice D.Y. Chandrachud entrenched this principle as one of the quintessential elements of the right to equality by observing:

> While assessing the reasonableness of an accommodation, regard must also be had to the benefit that the accommodation can have, not just for the disabled person concerned, but also for other disabled people similarly placed in future.

> As the Committee on the Rights of Persons with Disabilities noted in General Comment 6, reasonable accommodation is a component of the principle of inclusive equality. It is a substantive equality facilitator. (*Vikash Kumar v Union Public Service Commission* (2021), para. 48–49)

Thus, to conclude, the key lessons emerging from this discussion can be briefly stated as follows. Firstly, social action litigation has the prospect of producing results even in low-priority areas of constitution and legislation. Secondly, disability rights adjudication often transcends the traditional adversarial mode of litigation, marking the collaboration between public spirited individuals, disabled persons' organisations, and the State. The same has harnessed the constitutionalisation of the right to accessibility. This being an enabler right, its effective enforcement may go a long way to opening pathways for disabled persons to the enjoyment of other connected rights. Thirdly, the jurisprudence evolved by the courts regarding the interests and rights of disabled persons is the only silver lining around the pervasive constitutional and legislative deficit.

However, litigation will yield results only if jurisprudence is informed with the overarching principles of accessibility, respect for difference and human dignity underlying UNCRPD. The discussion also demonstrates that strategic litigation, instead of namesake litigation, is a way forward for the enforcement of the right to accessibility.

## 5. Limitations of the Enforcement of Rights to Accessibility in India

This section pulls the threads of the preceding discussion together by briefly identifying the limitations of the Indian approach to the enforcement of accessibility rights and offering ways forward. As has already been elaborated in Section 3, the primary objective underlying the Accessible India Campaign, RPwD Act 2016, and building laws is to ensure that all government programmes, such as the Smart City Mission, are proactive and sensitive to the needs of people with disabilities, in order to ensure their easy access to facilities such as public and private buildings, workplaces, commercial activities, public utilities, religious, cultural, leisure or recreational activities, health services, law enforcement agencies and transport infrastructure.

The intention behind such drives towards enhanced accessibility is apparent. These requirements and programmes are, however, not consistent with on-the-ground reali-

ties, because of the lack of effective interaction with the target groups and of efficacious enforcement machinery (Addlakha 2021; Jain and Jain 2024[14]).

Part of the problem also lies in the adoption of a very welfarist, paternalistic and instrumentalist approach to the realisation of accessibility. It is viewed primarily as an obligation confined to the provision of barrier-free access to public buildings and offices, without any sensitivity or attention to the right to walkability and the principle of mobility justice as catalysts for the full realisation of the principle of accessibility. Can someone access a public office or a building without any policies or programmes in place to make the roads and streets leading to these offices walkable, particularly for persons with disabilities? In other words, there is an integral connection between roads and streets and public offices and buildings, but the former are often overlooked.

Streets and roads not only facilitate navigation but shape our social interaction. In this sense, the landscape is but a taskscape (Holstein 2021). Streets and roads are meeting places for people, and much of the culture of mankind has been evolved in such public spaces. This ensures that, as the urban environment and its social configurations continuously evolve, the new skills and capabilities of all residents, including those with disabilities, are taken into account. Thus, it is incumbent to tailor the fundamental right "to move freely" to the special circumstances and situations of persons with disabilities.

In India, with huge pressure on limited resources and infrastructure from a variety of interests, the judiciary has played an important role in not only lending a voice to, but also providing the recognition due to persons with disabilities in the domain of accessibility. However, its role cannot be viewed as free from limitations.

The notion of judicial minimalism also influences the degree of enforcement by courts, as has been rightly highlighted: "...judges decide cases properly to the extent that they minimise their own imprint on the law by meticulously assessing 'one case at a time,' ruling on narrow and shallow grounds, eschewing broader theories, and altering entrenched legal practices only incrementally" (Smith 2010). In our opinion, along with a reliance on judicial enforcement—which currently seems to be the exclusive means through which disability rights are realised—there should be a greater focus on the initiation of campaigns and the creation of pressure groups by civil society, in order to arouse the conscience of the State and provoke it to foster a culture of inclusive and accessible public streets.

Further, in India, compensation is not perceived as an efficacious remedy for breaches of civil rights. Local authorities are not expected to be held liable for a breach of statutory duties by way of compensatory relief. The emphasis is instead on corrective measures and addressing the accountability of authorities by way of censure and allied actions. The gap between breaches of statutory duties relating to disabled persons on one hand, and enforcement on the other, produces undesirable outcomes. Recourse to the writ jurisdiction of the High Courts and the Supreme Court does not produce predictable results in all cases due to the summary procedures involved, as well as the expense and time-required. These courts, exercising writ jurisdiction, may not be in a position to investigate the violation of statutory rights and supervise the implementation of schemes, programmes and campaigns to be undertaken by the State. In fact, the Supreme Court of India has recently lamented the perfunctory approach taken in provinces across the country to the establishment and implementation of machinery envisaged under the RPwD Act 2016 for its effective enforcement (Jain 2022). At times, due to the difference of opinion between the High Court and the Supreme Court, petitions may also prove to be ineffective. For example, in *Nipun Malhotra v Government of NCT, Delhi (2017)*, the Delhi High Court provided salutary relief, but on appeal, the Supreme Court overlooked the demands of the petitioner (*Government of NCT of Delhi and others v Nipun Kumar Malhotra (2018)*). The approach of the Supreme Court demonstrates that when it comes to policy decisions, such as the allocation of resources (for example, through the procurement of low-floor buses), the courts are more reluctant

---

[14] Excerpted from the talk delivered by Sanjay Jain in COSP17 Side Event: Toward Inclusive and Accessible Pedestrian Environments: Implementation Challenges and Promising Practice from Around the World (June 2024).

to intervene than in cases in which the violation is of an individual's right to accessibility (Jain 2022).

Furthermore, in India, cases of pedestrians with protected characteristics can generally reach courts only if there is an active NGO or disabled persons' organisation to support them, mainly due to cost considerations. Considering the dominance of ableist design and the nascent evolution of the principle of accessibility in India, unsafe streets or street features simpliciter may not have sufficient legal purchase to raise tort claims. The same, however, when linked with additional claims like negligence, may be effective in recovering compensation, as demonstrated by the survey of cases in Section 4. Moreover, if the accessibility initiatives taken by the government comply with the standards prescribed by the experts, the courts are likely to be deferential. However, often, such opinions may not factor in the viewpoints of stakeholders.

To epitomize, the courts must wed the issue of unsafe street design to the fundamental right to move freely throughout India (guaranteed under Article 19(1) (d) of the Constitution) (Constitution of India 1950), counting on the ideals of constitutional morality (Grote 2023) and transformative constitutionalism (Klare 1998), to develop a public duty of government bodies to combat and interrogate the dominant discourse of ableism. The former entails paramount respect to the Constitution and the values underlying it, and the judges may tailor the same to the context of physical and mental disability by infusing constitutionalism with the principles of reasonable accommodation and accessibility inherent to the values of equality and dignity. Similarly, the latter obligates the courts to ensure and uphold the supremacy of the constitution, while at the same time mandating that a sense of transformation is ushered in society constantly and endlessly by interpreting and enforcing the constitution, as well as other provisions of law, in consonance with the avowed object. The court may fine-tune this principle in the domain of disability rights by inviting attention to the predominance of ableism and the need to transform ableist structures and the design of the legal and social order. The potential of these ideals in domains such as minority rights and governmental lawlessness further justifies the argument (*Indian Young Lawyers Assn. (Sabarimala Temple-5J.) v State of Kerala, (2019) 11 SCC 1: 2018 SCC OnLine SC 1690*; *Manoj Narula v Union of India, (2014) 9 SCC 1: 2014 SCC OnLine SC 640*).

However, at the same time, we wish to strike a cautionary note against over reliance on the idea of unenumerated rights (i.e., implied rights or rights which are not explicitly mentioned under the constitution). It would therefore be more plausible to recognise the right to accessibility as concomitant with the specific enumerated right to freedom of movement guaranteed by the Constitution of India. In this light, State action under the relevant provisions of the RPwD Act 2016, laying down accessibility obligations, would further strengthen and cement the principle of accessibility in the Indian legal order.

The need of the hour, therefore, is to adopt a bottom-up approach to the enforcement of the RPwD Act 2016 in general, and the right to accessibility in particular.

## 6. Conclusions

Lawson et al. have rightly observed that the unplanned and widespread expansion of almost all major cities across the globe has resulted in fragmentation, lack of walkability, and car dependency, thereby inhibiting the "democratic right to access the city and the city's public spaces" (Lawson et al. 2022). Therefore, road planning is the key to dispensing with excessive car dependency and to promoting uninhibited navigation and active travel, particularly for disabled persons, women and elderly persons. Additionally, to facilitate and foster accessibility for disabled persons, planners also need to address the problems faced by disabled persons in navigating environments without kerbs and interacting with vehicles.

Having introduced the theme of the paper in Section 1, in Section 2, we set out competing conceptions of accessibility. In Section 3, we considered how accessibility is legally entrenched in India by analysing relevant constitutional and statutory provisions. We also shed light on policies, campaigns and programmes initiated by the government for the enforcement of the principle of accessibility. In Section 4, we explored the adjudication

of the right to accessibility in India in detail, to demonstrate the role of judges as thought-leaders. In Section 5, we identified the limitations of the Indian approach to accessibility and offered brief insights to overcome the same. We thereby sought to investigate the role of the courts as a counterbalance to the majoritarian indifference to the concerns of persons with disabilities. We demonstrated that, against the deeply ableist framing of the Indian Constitution, the courts have found opportunities to entrench and infuse the principles of accessibility so as to foster the status of persons with disabilities as equal citizens. However, we question the extent to which mainstream public law and the activist stance of the Indian judiciary recognises the democratic deficit experienced by disabled persons as a sign of democratic dysfunction (Dixon 2023a).

This paper seeks to make a strong case for effective coordination between authorities to implement the mandate of the RPwD Act 2016, and to translate the spirit of Article 9 of the UNCRPD in policies and programs impacting accessibility. We have demonstrated how the Indian judiciary has been an important catalyst, along with disabled persons' organisations and public-spirited advocates, for pushing the cultivation of accessible and barrier-free environments, in which shared public spaces are more inclusive and friendly for disabled persons.

We have also demonstrated the existence of a gap between the pronouncement of orders and their actual implementation. In our opinion, the poor implementation of Sections 84 and 85 of the RPwD Act 2016, which obligate the State to establish special courts to try the offences they create and to appoint Special Public Prosecutors to conduct cases in such courts, has hampered the private law adjudication of disability rights and has also denied disabled people access to justice at local and district levels.

Although reliance on constitutional courts has provided an impetus for the evolution of the public law on disability rights, its evolution is subject to the limitation of the summary procedure of writ jurisdiction. Furthermore, persons with disabilities, for want of resources, have to depend largely on pro bono lawyering. In other words, in the absence of pro bono lawyers, many violations of the rights of disabled people may go unaddressed. With the ratification of UNCRPD by India, accessibility and mobility, the core of most human rights and fundamental freedoms for disabled persons, cannot be viewed as luxuries any longer.

**Author Contributions:** Conceptualisation, preparing original draft, review and editing, S.J. and M.J. All authors have read and agreed to the published version of the manuscript.

**Funding:** This research received no external funding.

**Institutional Review Board Statement:** Not applicable.

**Informed Consent Statement:** Not applicable.

**Data Availability Statement:** Data are contained within the article.

**Acknowledgments:** We gratefully acknowledge the inputs from Anna Lawson, University of Leeds, and Maria Orchard, Research Fellow, Inclusive Public Space project, School of Law, University of Leeds.

**Conflicts of Interest:** The authors declare no conflict of interest.

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
