# Peer review of "Revisiting the Conceptual Terrains of the Right to Accessibility in India: The Role of Judicial Enforcement"

_laws, 2024_

Round 1

Reviewer 1 Report

Comments and Suggestions for Authors

The theoretical framing by using Titchkosky is valuable but while this work is influential, integrating additional viewpoints could provide a more balanced and comprehensive analysis. Titchkosky's work primarily focuses on Western contexts. There might be cultural and contextual differences when applying her theories to the Indian legal framework. Are there challenges or adaptations needed when applying her theories to India? Are there authors from an Indian perspective that might enrich the theoretical and philosophical argument you present? 

Section I

Line 82: "positively purple" what does this mean? 

144: good contrast between transformative and ableist approaches to disability. 

170ff: very interesting discussion of the Indian federal system. 

Lines 208-219: The paper identifies judges who are to have a role in advocating for accessibility for people with disabilities. Is this the role of the judiciary in Indian law, or is this more the role of the legislators. Have you any concerns about judicial activism that might over extend their powers, even for worthy causes? 

The discussion on the scope of the Indian constitution, and the role of the legislator indicates a level of frustration with regard to disability rights not being realised, and a moral duty for the judiciary to take up the cause of disability rights. However, it may still not be the case that the discussion you offer, 'clearly establishes justification for interventionist and counter-majoritarian institutions, like constitutional courts' (line 265).

I think more care is required in this argument, because while it will satisfy the activist argument, this may well fail in terms of the constitutional balance of powers. I think it is important to reflect on the necessary constitutional checks on judicial overreach even for causes that are really worthy. 

line 320: the argument for a broad interpretation of the provisions of the constitution, is a challenging one and could be better contextualised from other constitutional law arguments who might disagree with such a position. 

Section II

The second part is a worthwhile discussion of the legislative acts in the context of the UNCRPD. Again this might be a little descriptive without a segment drawing out the consequences, and possibly reflecting back on the theoretical underpinnings by Titchkosky and others on the impact of these measures on people with disabilities.  

Section III

the case law is really interesting, and well worth undertaking this detailed exploration to examine both the effectiveness of the legislation and the judgements of the courts. With this in mind, it would be helpful to review these contrasting approaches and the scope or latitude that they are afforded while delivering rights for people with disabilities. 

Line 792 "dashed the deceased", not clear what this means?

Section III carries out what might be described as judicial case studies (streets, Universities, and public buildings) but what can we learn from these instances?

Line 944 goes some way to begin that reflection on the learning from the jurisprudence.

I think this concluding argument could be stronger, and given its own heading or section, to draw clear conclusions from the practice of the courts and the legislator, so as to draw out both the strengths and weaknesses of these two branches of the Indian state. There are definitely some good arguments here, and they are worth of development to explore their consequences.  Tying these learnings better together would be assisted by linking this section reflection to your theoretical framework by the likes of Titchkosky and others. From that point you could more clearly argue your case as to the way forward for disability rights in India. 

Finally, this is an informative and well written and argued paper, and also relevant to the non-Indian law specialist. A good window into Indian legal debate.

Thank you. 

Reviewer 2 Report

Comments and Suggestions for Authors

I thought this paper was clear and very well-structured, and the arguments compelling.  My main suggestion would be to move the discussion in the Conclusion into a prior section (a new section 4?) because you make some important substantive points about the limitations of the Indian approach to accessibility that deserve more than a perfunctory mention in the Conclusion, which I would recommend be short and summary in nature.

The quality of the writing is excellent, though there were some points where I thought the paper tended to use a bit too much jargon for my taste. More technically, there were a few places where the placement of commas was odd (e.g., p.24, line 944, after "Unlike"; earlier, after "Although") and I thought the capitalization in footnote 13 (p.26) was odd as well. At the end, "Case Laws" should be "Case Law."  Also, there were some coinages with which I (as an American English speaker) was unfamiliar--e.g., "conscientization" and "commucational" on p.14.  If these terms are not in common usage in your country I would consider using less recondite terminology.  I also was not sure what you meant by "positively purple" environment" (p.3, line 82).  On p.26, line 1016 "he" should be "the."

Comments on the Quality of English Language

 Minor editing of English language required

Round 2

Reviewer 1 Report

Comments and Suggestions for Authors

Thank you for the revisions of your paper, and making such thorough revisions to your work. While I would still argue that this paper is overly confident in the power of judicial activism in the Indian constitutional system, I believe this is an academic point and one that others could take up through this journal. I comment from a European public law perspective, and take the view that overt judicial activism can lead to problematic issues both for the judiciary and the legal system. My understanding of the Indian legal system, which is reflective of the European, is no different. 

I would only add that it would help to remove bullet points from your paper to make your case, and possibly edit down the overall final conclusions of your paper. This would involve some very minor changes on the overall read of your paper. 

Otherwise, your paper is interesting and contributes to the debate on the status of people with disabilities, and how they navigate the Indian legal system. Well done in engaging in this process. 

Author Response

  1. Summary

We are extremely grateful to both reviewers for their kind words about the article, their insightful comments and their helpful suggestions. Please find below a complete list of comments and our responses.

  1. Reviewer 1 Comment:

Thank you for the revisions of your paper and making such thorough revisions to your work. While I would still argue that this paper is overly confident in the power of judicial activism in the Indian constitutional system, I believe this is an academic point and one that others could take up through this journal. I comment from a European public law perspective and take the view that overt judicial activism can lead to problematic issues both for the judiciary and the legal system. My understanding of the Indian legal system, which is reflective of the European, is no different.

I would only add that it would help to remove bullet points from your paper to make your case, and possibly edit down the overall final conclusions of your paper. This would involve some very minor changes on the overall read of your paper.

Otherwise, your paper is interesting and contributes to the debate on the status of people with disabilities, and how they navigate the Indian legal system. Well done in engaging in this process.

  1. Response to Reviewer 1:

Thank you for the observations and kind appreciation. We have removed bullet points from the paper and have made appropriate edits to the conclusion in track change mode.

Reviewer 2 Report

Comments and Suggestions for Authors

On the attached, I have listed a number of small errata that are easily corrected, as well as a relatively few substantive comments that you might consider. One meta-suggestion: you describe the paper as limited to street accessibility but your discussion of cases and statutes is much broader than that. I would just describe the paper as addressing environmental accessibility (or a similar term) and not limit yourself to street access.

You have addressed well the concerns I expressed in my prior review.

Comments on the Quality of English Language

The quality of the English is excellent.  I would review the attached comments for additional places where you need to make edits.  There is much less jargon in this version of the paper.  I would be a bit careful regarding some of your phrases (e.g., roaring like a toothless tiger or glaring apathy) that, while not incorrect, are a bit too dramatic for the points you are making.

Author Response

  1. Reviewer 2 Comment 1:

On the attached, I have listed a number of small errata that are easily corrected, as well as a relatively few substantive comments that you might consider. One meta-suggestion: you describe the paper as limited to street accessibility but your discussion of cases and statutes is much broader than that. I would just describe the paper as addressing environmental accessibility (or a similar term) and not limit yourself to street access.

You have addressed well the concerns I expressed in my prior review.

  1. Response to Reviewer 2 Comment 1:

Thank you for the observation. We have addressed the meta-suggestion by making relevant changes (Lines 8, 23-24, and 748) and have also rectified the errata by fixing errors in the track changes mode.

We have indented the long quotations by 5cm from the left and 1cm from the right.

There were few comments like those pointed out at page 8, line 310 (current line 363), and page 12, line 433 (current line 519), which do not require any modification. We have done double check for the same.

Further, as regards the comment on page 17, lines 644-46 (current lines 777-779), the description in these lines is not inconsistent with the preceding paragraph.

Similarly, regarding the comment on page 23, lines 873-876 (lines 1028-1031), we are referring to the approach of the respondents as cavalier and not that of the court.

We have also removed footnotes 15 and 16 and shifted their content to the text. (Page 28)

There seems to be an issue with the footnote numbering as after footnote 9, directly there is footnote 11, i.e. footnote number 10 is missing. (Pages 12-13)

  1. Reviewer 2 Comment 2:

The quality of the English is excellent.  I would review the attached comments for additional places where you need to make edits.  There is much less jargon in this version of the paper.  I would be a bit careful regarding some of your phrases (e.g., roaring like a toothless tiger or glaring apathy) that, while not incorrect, are a bit too dramatic for the points you are making.

  1. Response to Reviewer 2 Comment 2:

Thank you for the observations and kind appreciation. We have made appropriate changes (lines 534 and 953).